# The number of cases, mortality and treatments of viral hemorrhagic fevers: A systematic review

**Drifa Belhadi**[1,2]*, **Majda El Baied**[2], **Guillaume Mulier**[2], **Denis Malvy**[3,4], **France Mentré**[1,2], **Cédric Laouénan**[1,2]

**1** Université Paris Cité, Inserm, IAME, Paris, France, **2** AP-HP, Hôpital Bichat, Département d'Épidémiologie Biostatistiques et Recherche Clinique, Paris, France, **3** UMR 1219 Inserm/EMR 271 IRD, University of Bordeaux, Bordeaux, France, **4** Department for Infectious and Tropical Diseases, University Hospital Center Pellegrin, Bordeaux, France

* drifa.belhadi@aphp.fr

## Abstract

### Background

Viral hemorrhagic fevers (VHFs) are a group of diseases, which can be endemo-epidemic in some areas of the world. Most of them are characterized by outbreaks, which occur irregularly and are hard to predict. Innovative medical countermeasures are to be evaluated but due to the field specificities of emerging VHF, challenges arise when implementing clinical studies. To assess the state of the art around VHFs, we conducted a systematic review for all reports and clinical studies that included specific results on number of cases, mortality and treatment of VHFs.

### Methods

The search was conducted in January 2020 based on PRISMA guidelines (PROSPERO CRD42020167306). We searched reports on the WHO and CDC websites, and publications in three international databases (MEDLINE, Embase and CENTRAL). Following the study selection process, qualitative and quantitative data were extracted from each included study. A narrative synthesis approach by each VHF was used. Descriptive statistics were conducted including world maps of cases number and case fatality rates (CFR); summary tables by VHF, country, time period and treatment studies.

### Results

We identified 141 WHO/CDC reports and 126 articles meeting the inclusion criteria. Most of the studies were published after 2010 (n = 97 for WHO/CDC reports and n = 93 for publications) and reported number of cases and/or CFRs (n = 141 WHO/CDC reports and n = 88 publications). Results varied greatly depending on the outbreak or cluster and across countries within each VHF. A total of 90 studies focused on Ebola virus disease (EVD). EVD outbreaks were reported in Africa, where Sierra Leone (14,124 cases; CFR = 28%) and Liberia (10,678 cases; CFR = 45%) reported the highest cases numbers, mainly due to the 2014–

**Data Availability Statement:** All relevant data are within the manuscript and its Supporting Information files.

**Funding:** The authors received no specific funding for this work.

**Competing interests:** The authors have declared that no competing interests exist.

2016 western Africa outbreak. Crimean-Congo hemorrhagic fever (CCHF) outbreaks were reported from 31 studies in Africa, Asia and Europe, where Turkey reported the highest cases number (6,538 cases; CFR = 5%) and Afghanistan the last outbreak in 2016/18 (293 cases; CFR = 43%).

Regarding the 38 studies reporting results on treatments, most of them were non-randomized studies (mainly retrospective or non-randomized comparative studies), and only 10 studies were randomized controlled trials. For several VHFs, no specific investigational therapeutic option with strong proof of effectiveness on mortality was identified.

## Conclusion

We observed that number of cases and CFR varied greatly across VHFs as well as across countries within each VHF. The number of studies on VHF treatments was very limited with very few randomized trials and no strong proof of effectiveness of treatment against most of the VHFs. Therefore, there is a high need of methodologically strong clinical trials conducted in the context of VHF.

## Author summary

Viral hemorrhagic fevers are a group of febrile illnesses that constitutes a challenge to public health. Viral hemorrhagic fevers are endemic in some areas of the world and are often associated with high morbidity and mortality. Due to high mortality rates and outbreaks that are irregular and hard to predict, difficulties arise when undertaking clinical studies to assess new treatments against viral hemorrhagic fevers. To provide an extensive overview of viral hemorrhagic fevers, we conducted a systematic review to retrieve available information on number of cases, mortality and treatments of viral hemorrhagic fevers. We found that cases number and mortality varied greatly across outbreaks and across countries within each viral hemorrhagic fever. The number of studies on viral hemorrhagic fever treatments was very limited with very few methodologically strong studies. Moreover, for several viral hemorrhagic fevers we did not identify specific investigational therapeutic option with strong proof of effectiveness on mortality. Therefore, there is a high need to conduct methodologically strong studies to investigate treatments against viral hemorrhagic fevers.

## Introduction

Viral hemorrhagic fevers (VHFs) are a group of febrile illnesses caused by four families of RNA viruses: arenaviridae, filoviridae, bunyaviridae and flaviviridae. [1] These highly infectious viruses are mainly zoonotic; meaning they naturally exist in animal or insect populations. [2] When a person encounter an infected animal or insect, the virus can spread through spillover into the human population, and subsequently is transmitted from person-to-person through contact with blood or other body fluids. Whatever their capacity to drive paramount hemorrhagic manifestations, many VHFs can cause severe, life-threatening disease. The agents that are causative of VHF are often classified as Biosafety Level 4 (BSL-4) pathogens that require special laboratory facilities with the highest level of safety measures. [2,3] VHFs are distributed worldwide and are often associated with high morbidity and mortality. Most of them

are characterized by clusters or even outbreaks occurring irregularly and almost resulting from spillover or more recently from human reservoirs constituted by immunologically preserved sanctuaries where the virus may persist after recovery of conversant survivors.

Patient outcomes are highly associated with the timing of curative treatment with improved outcomes when the specific or supportive therapy is started early. [4,5] Hence, VHF care remains essentially supportive and some VHFs are treated only with basic medical care that is not always reaching the optimized level of standards aimed to prevent or control the multi-systemic disorders that account for bad outcome. [2,5,6] Available VHF drugs are limited and clinical data on the efficacy of VHFs drugs is restricted. [4,5] New investigational treatments need to be evaluated but due to the field specificities of emerging VHF, difficulties arise when conducting clinical studies. Indeed, hard to predict outbreak duration leads to limited number of recruited patients. Moreover high case fatality rate (CFR) leads to reluctance to use methodologically strong trial design such as randomized controlled trials as part of the patients will not receive the potentially beneficial treatment. [7]

VHFs have recently caused various outbreaks around the world. To assess the state of the art around VHFs, we systematically reviewed the World Health Organization (WHO) and Centers for Disease Control (CDC) websites and published literature for all reports and clinical studies that included specific results on number of cases, mortality and treatments of VHFs. We decided to focus mostly on VHFs caused by a selection of arenaviridae, filoviridae, bunyaviridae and flaviviridae and did not look at some other important VHF conditions such as the severe dengue and yellow fever.

## Methods

The systematic review was registered on the International prospective register of systematic reviews (PROSPERO 2020 CRD 42020167306). The objective of the systematic review was to review the case fatality rates, number of cases and treatment options of VHFs. The initial protocol also included the review of sequelaes, which will not be presented here.

### Eligibility criteria

The PICOS (Participants, Intervention, Comparison, Outcomes and Study types) framework was used to identify relevant data.

**Participants.** Humans infected with a pathogen causative of a VHF from the following list: [8]

- Alkhurma hemorrhagic fever (AHF)

- Argentine hemorrhagic fever (ArHF)

- Bolivian hemorrhagic fever (BHF)

- Chapare hemorrhagic fever (CHF)

- Crimean-Congo hemorrhagic fever (CCHF)

- Ebola virus disease (EVD)

- Hantavirus pulmonary syndrome (HPS)

- Hemorrhagic fever with renal syndrome (HFRS)

- Kyasanur Forest disease (KFD)

- Lassa fever (LF)

- Lujo hemorrhagic fever (LHF)

- Lymphocytic choriomeningitis (LCM)

- Marburg virus disease (MVD)

- Omsk hemorrhagic fever (OHF)

- Rift Valley fever (RVF)

- Sabia-associated hemorrhagic fever (SHF)

- Tick-borne encephalitis (from a hemorrhagic variant)

- Venezuelan hemorrhagic fever (VeHF)

**Interventions/Comparisons.** This review did not focus on any specific intervention.

**Outcomes.** Any studies reporting number of cases and/or CFR and/or mortality rates associated with treatments of a VHF were included.

**Study types.** Official information from WHO/CDC and national health websites, cross-sectional, cohort/case-control studies, descriptive reports and clinical trials were included.

**Study language.** Included studies were limited to studies published in English or French. The following types of study were deemed ineligible; case reports, case series, systematic reviews and meta-analyses.

## Search strategy

The systematic review consisted in two parts. The first search was conducted in December 2019 on the WHO and CDC websites to identify the number of cases and deaths associated with each disease by year and by country. The second search consisted in a systematic literature search of bibliographic databases based on the PRISMA guideline. Relevant studies were identified by searching MEDLINE, Embase and the Cochrane Central Register of Controlled Trials (CENTRAL) in the Cochrane Library. We searched the electronic databases until January 21$^{st}$, 2020. Search terms were developed using a combination of MeSH/EMTREE terms and free-text terms to capture the relevant populations, outcomes and study types (cf. S1 Fig in Supplementary Material). Reference lists of included studies were not reviewed. Additional hand searches were performed on national health websites.

## Study selection

For the systematic literature search, after removing duplicates, two researchers (DB and GM) independently evaluated all identified citations based on titles and/or abstracts. In case of disagreement, a third researcher (CL) served as tiebreaker. Full-text publications of studies included based on title and abstract were retrieved and reviewed by two researchers (DB and ME) to assess eligibility based on the inclusion/exclusion criteria. A third researcher (CL) served as a tiebreaker for any discordant decisions. Justification for study exclusion was documented.

## Data extraction

Information identified from the WHO and CDC websites were extracted for each disease by period and by country. For the systematic literature search, following the study selection process, qualitative and quantitative data were extracted from each of the included articles. Numbers were extracted as reported in each study. If the CFR was not reported in a study, the number of cases and the number of deaths were used, if available, to estimate the missing CFR.

### Risk of bias assessment

Quality assessment was performed using the Newcastle-Ottawa Quality Assessment Form [9] to evaluate cohort studies, and the Risk of Bias (RoB) tool described in the Cochrane Handbook for Systematic Reviews of Interventions [10] to evaluate randomized clinical trials.

### Strategy for data synthesis

A narrative synthesis approach by disease was used for each outcome. Descriptive statistics were conducted to describe the published articles and WHO/CDC reports. World map of the total number of cases and CFR were produced and summary tables of the number of cases and CFR by VHF, country and time period were also produced. Summary tables reporting the characteristics and results of each study on VHF treatments were produced.

World maps were produced using R software version 3.6.0.

## Results

A total of 57 reports were identified from the WHO website, and 96 reports from the CDC website. After removing duplicates, 141 reports were extracted. After removing duplicates, 4 461 publications were identified from Medline, Embase, CENTRAL and through hand searches. We excluded 4 153 publications based on the screening of titles and abstracts. A total of 308 publications were included for full text review. Finally, 126 publications met the inclusion criteria and were extracted. The selection process and the numbers at each stage are shown in Fig 1. The exhaustive lists of included publications and WHO and CDC reports are reported in S1 and S2 Tables in Supplementary Material.

### Characteristics of included studies

The characteristics of the included studies are summarized in Table 1. Most of the studies were published after 2010 (69%, n = 97 for WHO and CDC reports and 74%, n = 93 for publications) and were conducted mainly in Africa (57%, n = 81 for WHO and CDC reports and

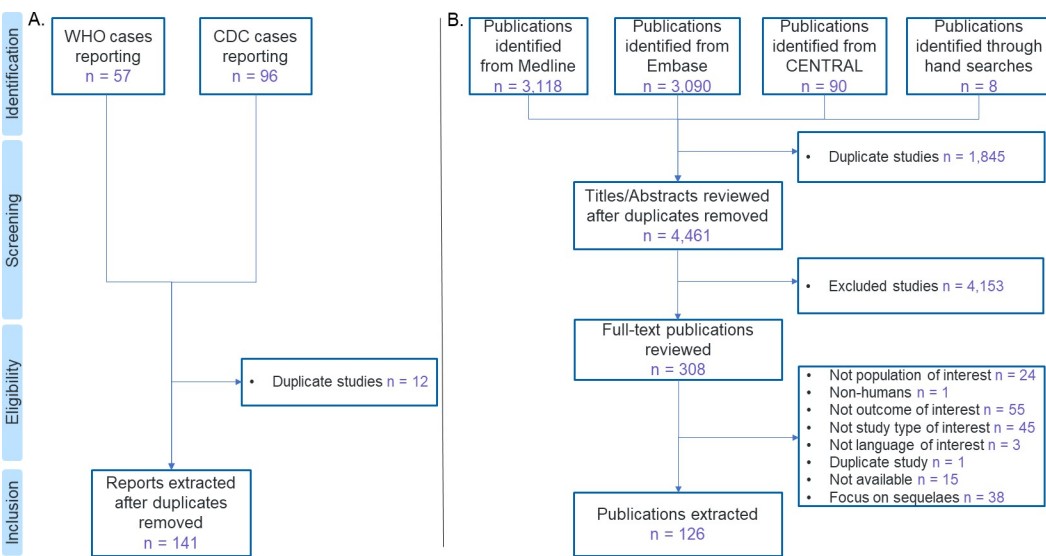

**Fig 1.** Study selection flow diagram: A. For the WHO and CDC websites search; B. For the systematic literature search.

48%, n = 60 for publications). The majority of studies reporting number of cases and/or CFR were descriptive reports (100%, n = 141 for WHO and CDC reports and 90%, n = 79 for publications) followed by retrospective studies (10%, n = 9 for publications). Regarding studies reporting results on treatments, 32% (n = 12) of them were retrospective studies, followed by 26% (n = 10) of randomized controlled trial and 16% (n = 6) of non-randomized comparative studies. Most of WHO and CDC reports focused on EVD (33%, n = 47), followed by HPS (18%, n = 26), LF (17%, n = 24) and RVF (16%, n = 23). In terms of publications, most of them focused also on EVD (33%, n = 43), followed by CCHF (21%, n = 27), HFRS (14%, n = 18) and HPS (10%, n = 13). No relevant data were identified for KFD, LCM and tick-borne encephalitis (from a hemorrhagic variant).

## Risk of bias of included studies

On the 38 studies reporting results on treatments, 10 were randomized trials and 28 were non-randomized. Regarding the randomized comparative studies, only two were classified as having a "low risk of bias", both published after 2010, and the other eight studies were classified as having "some concerns" (cf. S3 Table). Regarding the non-randomized studies, 13 were classified as "poor quality", 5 as "fair quality" and 10 as "good quality" (cf. S4 Table). On those 28 non-randomized studies, 6 were published before 2010 and 67% of them (n = 4/6) were classified as "poor quality", compared with 41% (n = 9/22) for those published after 2010.

The 88 studies reporting number of cases and CFR were evaluated using the Newcastle-Ottawa Quality Assessment Form: 59 were classified as "poor quality" and 29 as "fair quality" (cf. S5 Table). On these 88 studies, 31 were published before 2010 and a similar proportion than those published after 2010 were classified as "poor quality" (65%, n = 20/31, for studies before 2010, compared with 68%, n = 45/66, for studies published after 2010).

## Number of cases and CFR

The worldwide distribution of VHFs by country is reported in Fig 2. We identified studies reporting outbreaks of at least one VHF across 55 countries. We found studies on 4 different VHFs in South Africa: CCHF, RVF, MVD and LHF. Based on identified studies, a total of 16 countries were associated with 2 VHFs, mainly in Africa with 9 countries, followed by America with 3 countries and Asia and Europe with 2 countries each. More detailed results are reported below for the VHFs with at least 5 studies. They are presented by alphabetical order.

## Crimean-Congo hemorrhagic fever (CCHF)

The distribution of CCHF cases and CRFs are reported in Fig 3. Cases were reported in Africa, Asia and Europe in the following countries: Afghanistan, Bulgaria, Georgia, Iran, Iraq, Kazakhstan, Kosovo, Mauritania, Oman, Pakistan, South Africa, Tajikistan, Turkey and Uzbekistan. The first documented cases were reported in 1948–1969 in Kazakhstan (89 cases, CFR = 25%). At the date of the review, the country with the highest reported number of cases was Turkey with a total of 6,538 cases (CFR = 5%) and the last documented cases reported in 2016–2018 in Afghanistan (293 cases, CFR = 43%). More details by country and period are reported in S6 Table.

## Ebola virus disease (EVD)

The distribution of EVD cases and CRFs are reported in Fig 4. Outbreaks were reported in sub-saharan Africa in the following countries: Democratic Republic of the Congo, Gabon, Republic of Guinea, Liberia, Mali, Nigeria, Sierra Leone, Sudan and Uganda. The first reported

**Table 1.  Characteristics of included studies.**

| | WHO/CDC reports (N = 141) | Publications (N = 126) |
|---|---|---|
| **Year of publication** | | |
| Before 2000 | 23 (16%) | 13 (10%) |
| Between 2000 and 2010 | 21 (15%) | 20 (16%) |
| After 2010 | 97 (69%) | 93 (74%) |
| **Region of interest** | | |
| Africa | 81 (57%) | 60 (48%) |
| America | 33 (23%) | 18 (14%) |
| Asia | 11 (8%) | 40 (32%) |
| Asia and Europe | - | 1 (1%) |
| Europe | 16 (11%) | 6 (5%) |
| Worldwide | - | 1 (1%) |
| **Study type** | | |
| *Studies reporting number of cases and/or CFR* | *N = 141* | *N = 88* |
| Descriptive report | 141 (100%) | 79 (90%) |
| Retrospective study | - | 9 (10%) |
| *Studies reporting results on treatments* | | *N = 38* |
| Case-Control Study | - | 2 (5%) |
| Descriptive report | - | 3 (8%) |
| Non-randomized comparative study | - | 6 (16%) |
| Prospective and retrospective study | - | 1 (3%) |
| Randomized Controlled trial | - | 10 (26%) |
| Retrospective study | - | 12 (32%) |
| Single-arm trial | - | 4 (10%) |
| **Disease of interest (alphabetical order)** | | |
| Alkhurma hemorrhagic fever (AHF) | 1 (1%) | 2 (2%) |
| Argentine hemorrhagic fever (ArHF) | 0 (0%) | 3 (2%) |
| Bolivian hemorrhagic fever (BHF) | 0 (0%) | 1 (1%) |
| Chapare hemorrhagic fever (CHF) | 2 (1%) | 0 (0%) |
| Crimean-Congo hemorrhagic fever (CCHF) | 4 (3%) | 27 (21%) |
| Ebola Virus Disease (EVD) | 47 (33%) | 43 (33%) |
| Hantavirus Pulmonary Syndrome (HPS) | 26 (18%) | 13 (10%) |
| Hemorrhagic fever with renal syndrome (HFRS) | 0 (0%) | 18 (14%) |
| Kyasanur Forest Disease (KFD) | 0 (0%) | 0 (0%) |
| Lassa fever (LF) | 24 (17%) | 7 (5%) |
| Lujo hemorrhagic fever (LHF) | 1 (1%) | 1 (1%) |
| Lymphocytic choriomeningitis (LCM) | 0 (0%) | 0 (0%) |
| Marburg virus disease (MVD) | 13 (9%) | 1 (1%) |
| Omsk hemorrhagic fever (OHF) | 0 (0%) | 1 (1%) |
| Rift Valley fever (RVF) | 23 (16%) | 9 (7%) |
| Sabia-associated hemorrhagic fever/Sabia hemorrhagic fever (SHF) | 0 (0%) | 1 (1%) |
| Tick-borne Encephalitis (from a hemorrhagic variant) (TBE) | 0 (0%) | 0 (0%) |
| Venezuelan hemorrhagic fever (VeHF) | 0 (0%) | 2 (2%) |

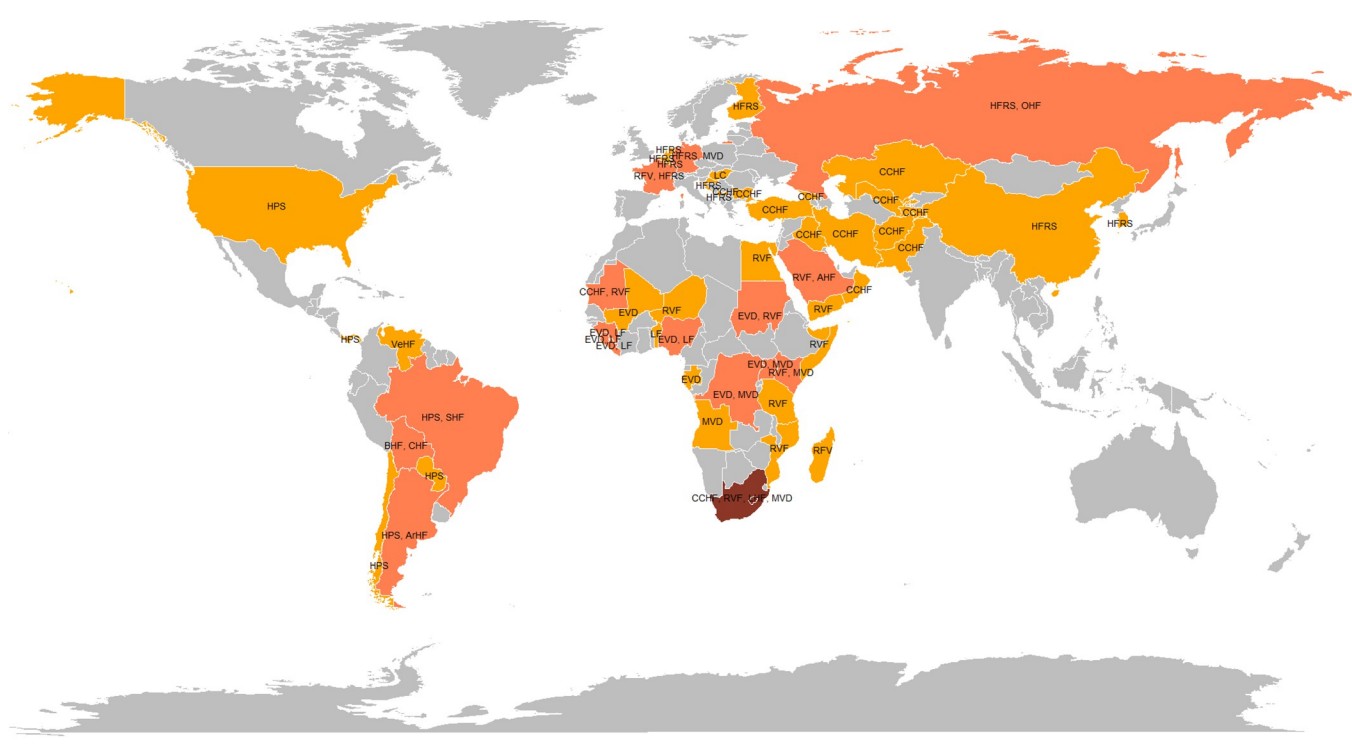

**Fig 2. Worldwide distribution of VHF.** The color denotes the number of VHF with at least one outbreak reported by country and the acronyms denote the names of the VHFs. Note: AHF, Alkhurma hemorrhagic fever; ArHF, Argentine hemorrhagic fever; BHF, Bolivian hemorrhagic fever; CHF, Chapare hemorrhagic fever; CCHF, Crimean-Congo hemorrhagic fever; EVD, Ebola Virus Disease; HPS, Hantavirus Pulmonary Syndrome; HFRS, Hemorrhagic fever with renal syndrome; LF, Lassa fever; LHF, Lujo hemorrhagic fever; MVD, Marburg virus disease; OHF, Omsk hemorrhagic fever; RVF, Rift Valley fever; SHF, Sabia hemorrhagic fever; VeHF, Venezuelan hemorrhagic fever. Maps were generated with the 'maps' R package using data from the Natural Earth Project (the 1:50m resolution version).

outbreak took place in 1976 in the Democratic Republic of the Congo (318 cases, CFR = 88%) and Sudan (284 cases, CFR = 53%). At the date of the review, the countries with the highest reported number of cases were Sierra Leone with a total of 14,124 cases (CFR = 28%) and Liberia with 10,678 cases (CFR = 45%), due to the 2014–2016 western Africa outbreak.

The reported CFRs since 2010 varied greatly across countries (cf. S7 Table). In total, 28% in Sierra Leone in 2014–2016 (14,124 cases), 40% in Nigeria in 2014 (20 cases), 41% in Uganda in 2012 (17 cases), 45% in Liberia (10,678 cases), 66% in the Democratic Republic of the Congo in 2018 (3,470 cases), 67% in Guinea in 2014–2016 (3,811 cases) and 75% in Mali in 2014 (8 cases).

## Hantavirus pulmonary syndrome (HPS)

The distribution of HPS cases and CRFs are reported in Fig 5. Cases were reported in America in the following countries: Argentina, Brazil, Chile, Panama, Paraguay and USA. The first documented cases were reported in 1993 in the USA (48 cases, CFR = 56%) and Brazil (884 cases from 1993 to 2006, CFR = 39%). At the date of the review, the countries with the highest documented number of cases were Brazil with 2,370 cases (CFR = 39%) and the last documented cases were reported in 2018 in USA (3 cases, CFR = 67%). More details by country and period are reported in S8 Table.

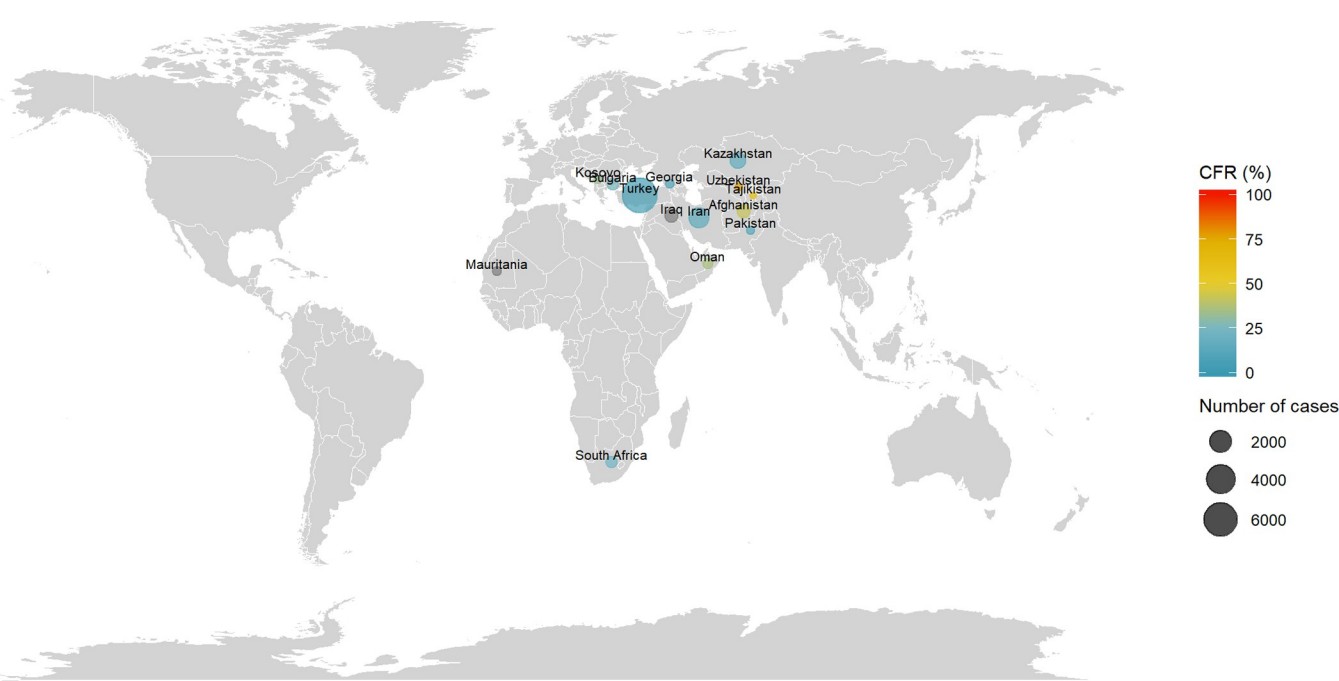

**Fig 3. Distribution of number of cases and CFR of Crimean-Congo hemorrhagic fever.** The size of the bubble is proportional with the number of cases reported and the color denotes the level of CFR by country. Note: Maps were generated with the 'maps' R package using data from the Natural Earth Project (the 1:50m resolution version).

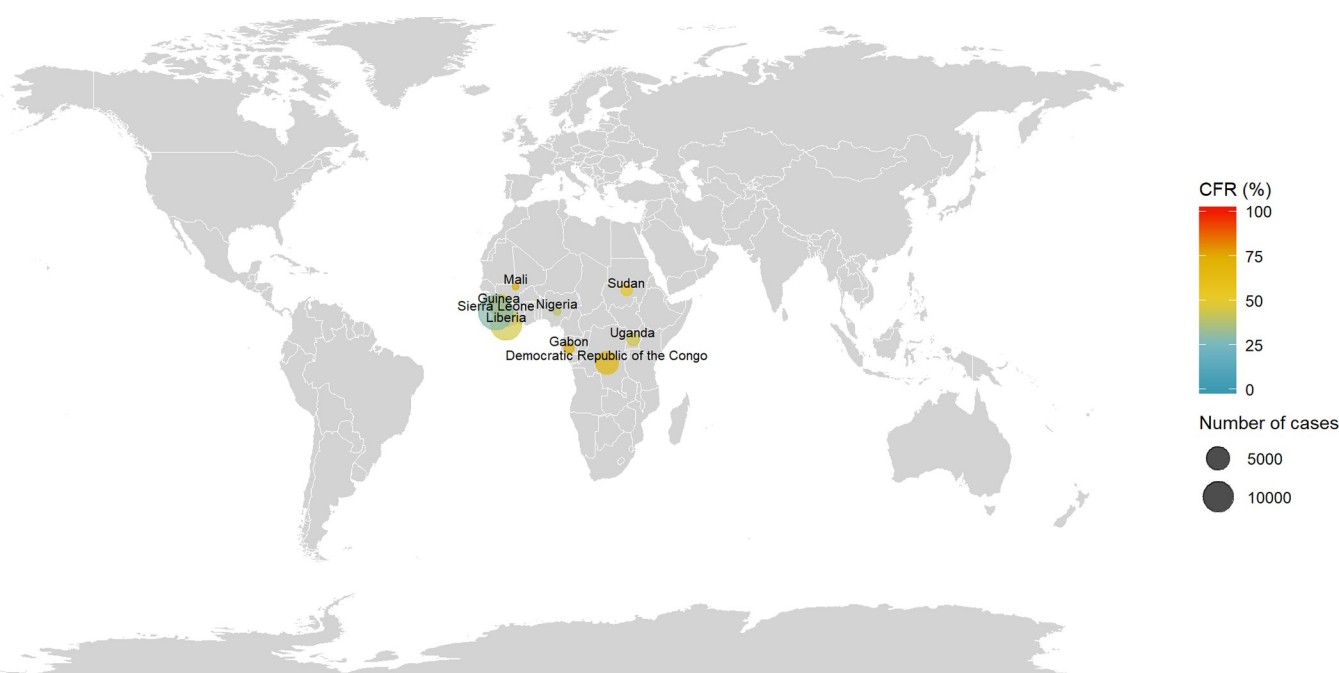

**Fig 4. Distribution of number of cases and CFR of Ebola Virus Disease.** The size of the bubble is proportional with the number of cases reported and the color denotes the level of CFR by country. Note: Maps were generated with the 'maps' R package using data from the Natural Earth Project (the 1:50m resolution version).

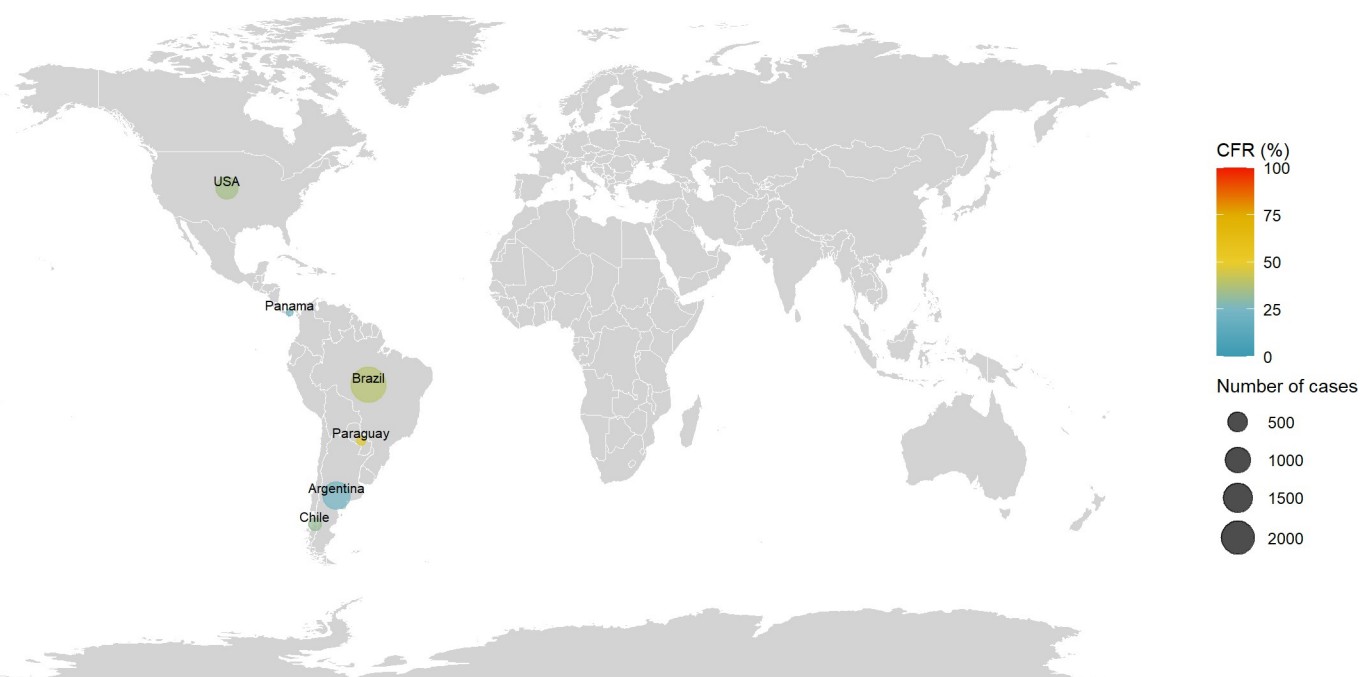

**Fig 5. Distribution of number of cases and CFR of Hantavirus Pulmonary Syndrome.** The size of the bubble is proportional with the number of cases reported and the color denotes the level of CFR by country. Note: Maps were generated with the 'maps' R package using data from the Natural Earth Project (the 1:50m resolution version).

## Hemorrhagic fever with renal syndrome (HFRS)

HFRS cases were reported in Asia and Europe in the following countries: China, Croatia, Finland, Belgium, France, Germany, Netherlands, Luxembourg, South Korea, Montenegro and Russia. The first documented cases were reported in 1931–1941 in China (10,000 cases, CFR = 30%). At the date of the review, the country with the highest reported number of cases was China with 1,306,812 cases (CFR = 3%) and the last documented cases reported in 2000–2017 in Russia (131,590 cases, CFR = 0.4%). More details by country and period are reported in S9 Table.

## Lassa fever (LF)

The distribution of LF cases and CRFs are reported in Fig 6. Cases were reported in western Africa in the following countries: Benin, Liberia, Nigeria, Sierra Leone and Guinea. We identified a study which reported the first outbreak in 1996–1999 in Guinea (22 cases, CFR = 18%). At the date of the review, the country with the highest reported number of cases was Nigeria with 2,287 cases (CFR = 23%) and the last documented cases reported in 2019 in South-West Nigeria (554 cases, CFR = 22%). More details by country and period are reported in S10 Table.

## Marburg virus disease (MVD)

MVD cases were reported in Africa and Europe in the following countries: Angola, Democratic Republic of the Congo, Germany, ex-Yugoslavia, South Africa, Kenya and Uganda. The first documented cases were reported in 1967 in Germany (29 cases, CFR = 24%) and ex-Yugoslavia (2 cases, CFR = 0%) and relied to lab.-accidental transmission in settings that were used to import monkeys from Central Africa. At the date of the review, the country with the highest reported number of cases was Angola with 374 cases (CFR = 88%) and the last

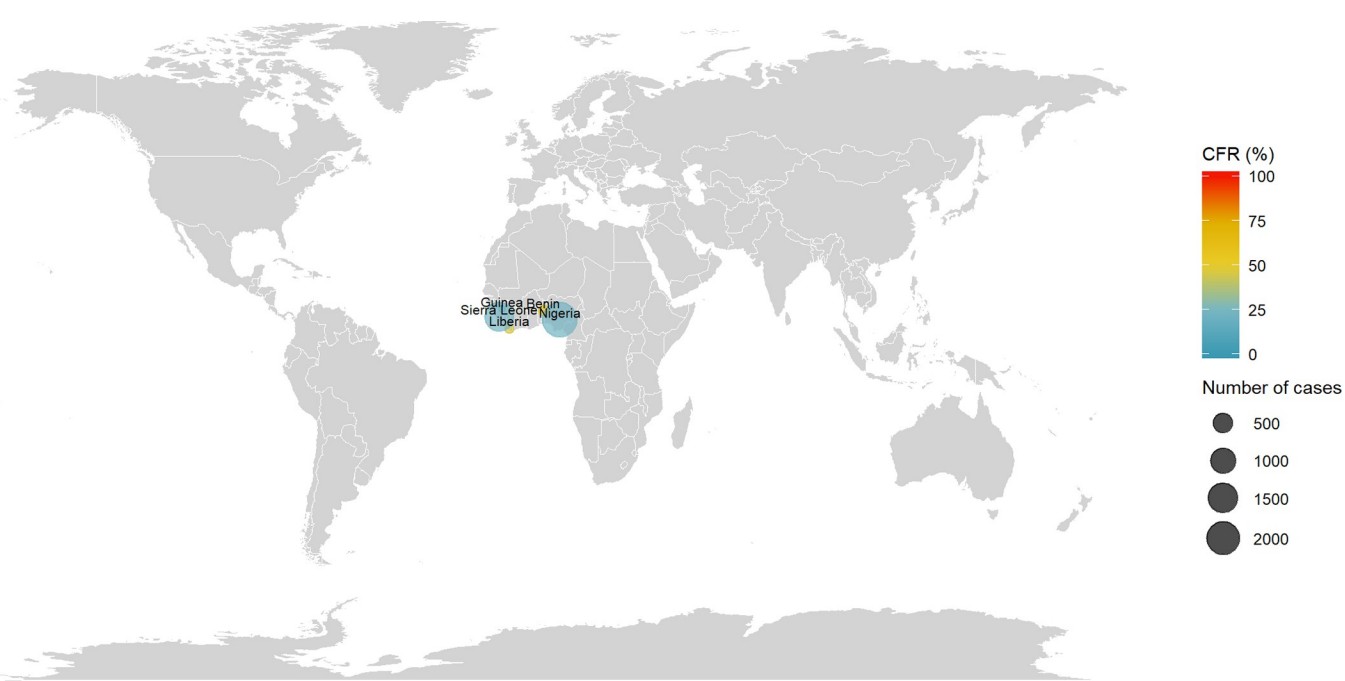

**Fig 6. Distribution of number of cases and CFR of Lassa fever.** The size of the bubble is proportional with the number of cases reported and the color denotes the level of CFR by country. Note: Maps were generated with the 'maps' R package using data from the Natural Earth Project (the 1:50m resolution version).

documented cases reported in 2017 in Uganda (2 cases, CFR = 100%). More details by country and period are reported in S11 Table.

## Rift Valley fever (RVF)

The distribution of RVF cases and CRFs are reported in Fig 7. Cases were reported in Africa and Asia in the following countries: Egypt, Kenya, Madagascar, Mauritania, Mayotte (France), Mozambique, Niger, Saudi Arabia, Somalia, South Africa, Sudan, Tanzania and Yemen. The first documented cases were reported in 1977–1978 in Egypt (18,000 cases, CFR = 3%). At the date of the review, the country with the highest documented number of cases was Egypt with 18,148 cases (CFR = 3%) and the last documented cases reported in 2018–2019 in Mayotte (129 cases, CFR not reported). More details by country and period are reported in S12 Table.

## Other VHFs

The remaining VHFs reported cases in one country each: Saudi Arabia for AHF (335 cases, CFR = 2%), Argentina for ArHF (981 cases, CFR not reported), Bolivia for BHF (690 cases, CFR = 23%), South Africa for LHF (5 cases, CFR = 80%), Russia for OHF (1144 cases, CFR = 14%), Brazil for SHF (4 cases, CFR = 2%) and Venezuela for VeHF (728 cases, CFR = 23%). More details by VHF and period are reported in S13 Table.

## VHFs treatments

Publications evaluating the efficacy of specific treatments on mortality were identified for Argentine hemorrhagic fever, Crimean-Congo hemorrhagic fever, Ebola Virus Disease, Hantavirus Pulmonary Syndrome, Hemorrhagic fever with renal syndrome and Lassa fever (cf.

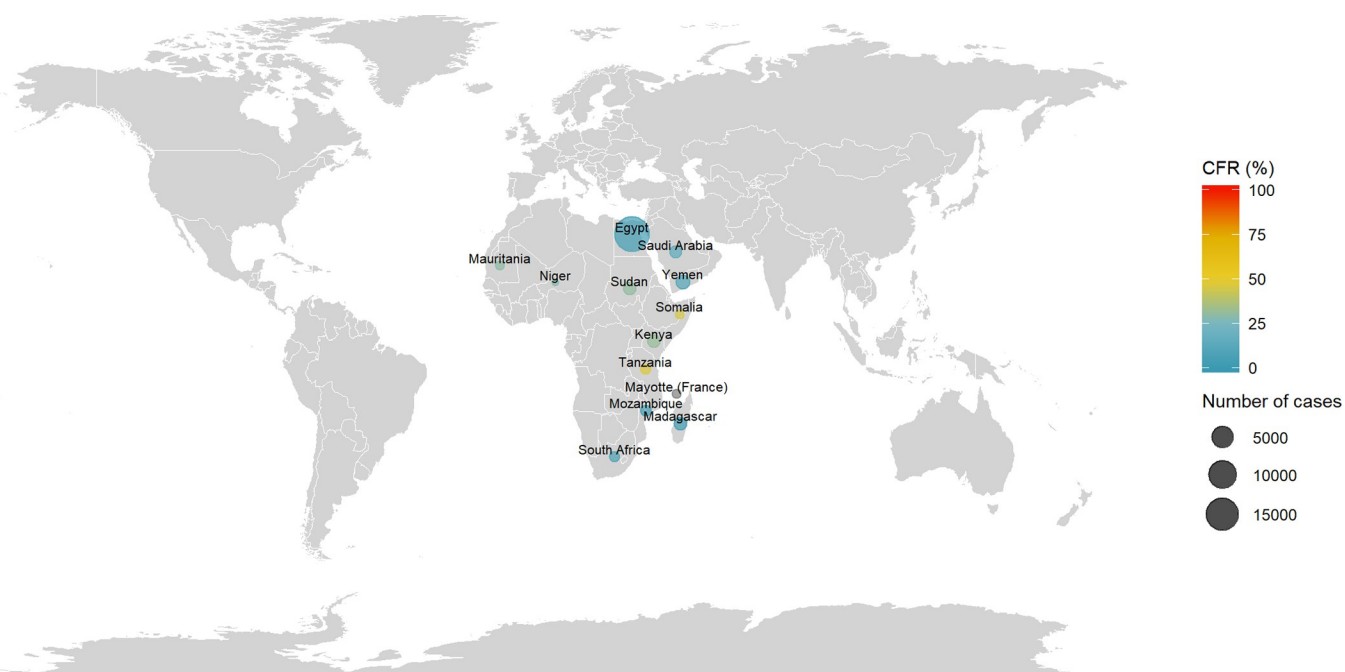

**Fig 7. Distribution of number of cases and CFR of Rift Valley Fever.** The size of the bubble is proportional with the number of cases reported and the color denotes the level of CFR by country. Note: Maps were generated with the 'maps' R package using data from the Natural Earth Project (the 1:50m resolution version).

Tables 2 and 3). More details on each investigational treatment are reported in S14 Table. Detailed results are reported below for the VHFs with at least one study. They are presented by alphabetical order.

## Argentine hemorrhagic fever (ArHF)

One study on treatment was identified for ArHF: a randomized controlled trial published in 1979 assessing intravenous immune plasma obtained from convalescent donors.[11] This trial showed a significantly decrease in mortality when immune plasma is given before the ninth day of the disease (CFR = 1.1%) compared with normal plasma obtained from donors without a history of ArHF (CFR = 16.5%).

## Crimean-Congo hemorrhagic fever (CCHF)

Nine studies on treatments were identified for CCHF: two randomized controlled trials, two non-randomized comparative studies, two case-control studies, two retrospective and/or prospective studies and one descriptive report. Most of the trials assessed oral or intravenous ribavirin, one trial assessed ribavirin +/- corticosteroids and one trial assessed immune globulins + ribavirin. Ribavirin was associated with inconsistent results across studies.

Two studies reported significant results of ribavirin and ribavirin +/- corticosteroids on mortality. However, those two studies had a relatively weak design (a non-randomized comparative study [19] and a prospective and retrospective study [13]). In the randomized trial comparing ribavirin versus standard therapy alone [18], ribavirin was associated with no positive effect on mortality (CFR = 6.3% versus 5.6%). The other randomized trial was a small study (40 patients, [20]) assessing polyvalent immune globulins + ribavirin which showed with no positive effect of the treatment on mortality compared with ribavirin alone (CFR = 25% versus 11%). Therefore, no strong proof of effectiveness of specific treatment against CCHF was identified.

**Table 2. Characteristics of included studies reporting results on treatments by VHF.**

| Study | Study design | Randomisation | Blinding | Country | Treatment | Comparator |
|---|---|---|---|---|---|---|
| **Argentine hemorrhagic fever** | | | | | | |
| **Maiztegui 1979** [11] | Randomized Controlled trial | Yes | Double-blind | Argentina | Immune plasma | Normal plasma obtained from donors without a history of AHF who were residents of Buenos Aires, a city located outside the endemic area of the disease. |
| **Crimean-Congo hemorrhagic fever** | | | | | | |
| **Elaldi 2009** [12] | Non-randomized comparative study | No | NA | Turkey | Ribavirin + Supportive therapy | Supportive therapy (patients who were diagnosed in 2003) |
| **Dokuzoguz 2013** [13] | Prospective and retrospective study | No | NA | Turkey | Ribavirin +/- Corticosteroid | No ribavirin +/- corticosteroid |
| **Cevik 2008** [14] | Descriptive report | No | NA | Turkey | Ribavirin + Supportive therapy | Supportive therapy |
| **Yilmaz 2016** [15] | Retrospective study | No | NA | Turkey and Iran | Ribavirin + Supportive therapy | Supportive therapy |
| **Izadi 2009** [16] | Case-Control Study | No | NA | Iran | Ribavirin + Supportive therapy | NR |
| **Tulek 2012** [17] | Case-Control Study | No | NA | Turkey | Ribavirin + Supportive therapy | Supportive therapy (patients with CCHF of the other infectious diseases department in the same hospital) |
| **Koksal 2010** [18] | Randomized Controlled trial | Yes | NR | Turkey | Ribavirin + Supportive therapy | Supportive therapy |
| **Mardani 2003** [19] | Non-randomized comparative study | No | NA | Iran | Ribavirin | No ribavirin (historical controls) |
| **Salehi 2013** [20] | Randomized Controlled trial | Yes | Single-blind | Iran | Immune globulins (IGIV) + ribavirin + Supportive therapy | Ribavirin + Supportive therapy |
| **Ebola Virus Disease** | | | | | | |
| **Aluisio 2019** [21] | Retrospective study | No | NA | Liberia and Sierra Leone | Intravenous fluid (IVF) + Supportive therapy | NR |
| **Aluisio 2019** [22]; **Aluisio 2019** [23] | Retrospective study | No | NA | Liberia and Sierra Leone | Vitamin A supplementation + Supportive therapy | Supportive therapy |
| **Aluisio 2020** [24] | Retrospective study | No | NA | Liberia and Sierra Leone | Cephalosporin + Supportive therapy | Supportive therapy |
| **Bai 2016** [25] | Retrospective study | No | NA | Sierra Leone | Favipiravir + Supportive therapy | Supportive therapy |
| **Sissoko 2016** [26] | Single-arm trial | No | NA | Guinea | Favipiravir + Supportive therapy | Historical data (540 patients hospitalized between 15 September and 15 December 2014, the 3 months preceding the start of the JIKI study) |
| **Yam 2020** [27] | Retrospective study | No | NA | Sierra Leone and Liberia | Multivitamin supplementation + Supportive therapy | Supportive therapy |
| **Dunning 2016** [28] | Single-arm trial | No | NA | Liberia | Brincidofovir + Supportive therapy | NR |
| **Gignoux 2015** [29] | Retrospective study | No | NA | Liberia | • Artemether-Lumefantrine + Supportive therapy <br><br> • Artesunate-Amodiaquine + Supportive therapy | Supportive therapy |
| **Garbern 2019** [30] | Retrospective study | No | NA | Liberia and Sierra Leone | Artesunate-amodiaquine (ASAQ) + Supportive therapy | Supportive therapy |

*(Continued)*

**Table 2.** (Continued)

| Study | Study design | Randomisation | Blinding | Country | Treatment | Comparator |
|-------|-------------|---------------|----------|---------|-----------|------------|
| **Dunning 2016** [31] | Single-arm trial | No | NA | Sierra Leone | TKM-130803 + Supportive therapy | NR |
| **Sahr 2017** [32] | Non-randomized comparative study | No | NA | Sierra Leone | Convalescent whole blood (CWB) + Supportive therapy | Supportive therapy |
| **Konde 2017** [33] | Single-arm trial | NA | NA | Guinea | IFN-beta 1a + Supportive therapy | Supportive therapy (Historical cohort). The historical control patients were admitted to the Coyah ETU during the same time period as the IFN beta-1a patients with RT-PCR confirmed blood EBOV. Also included 17 patients who matched the IFN treated patients for eligibility criteria based on <6 days from symptom onset, age, under care in a Guinean treatment centre, who were better matched for baseline CT values. |
| **Sadek 1999** [34] | Descriptive report | NA | NA | Democratic Republic of the Congo | Whole blood transfusion from convalescent patients + Supportive therapy | NA |
| **Van Griensven 2016** [35] | Non-randomized comparative study | No | NA | Guinea | Transfusion of convalescent plasma + Supportive therapy | Supportive therapy: Patients who had been admitted to the ETU during the preparatory period of the study while the system for apheresis and pathogen reduction was being set up and those for whom ABO compatible convalescent plasma was not available during the study. At the start of recruitment, there was a sufficient amount of convalescent plasma available to treat all the patients, so a protocol amendment was approved for the control group to consist of patients who were treated at the same ETU before the initiation of the trial. |
| **Davey 2016** [36] | Randomized Controlled trial | Yes | No | Liberia, Sierra Leone, Guinea and USA | ZMapp + Supportive therapy | Supportive therapy |
| **Mulangu 2019** [37] | Randomized Controlled trial | Yes | NA | Democratic Republic of the Congo | •Remdesivir + Supportive therapy<br>•Mab114 + Supportive therapy<br>•REGN-EB3 + Supportive therapy | ZMapp (intravenous 50mg per kilogram of body weight every third day beginning on day 1 for a total of 3 doses) + Supportive therapy |
| **Kerber 2019** [38] | Retrospective study | NA | NA | Guinea | Favipiravir + Supportive therapy | Supportive therapy |
| **Hantavirus Pulmonary Syndrome** | | | | | | |
| **Chapman 1999** [39] | Non-randomized comparative study | No | Open-label | United States | Ribavirin | No ribavirin (contemporaneous patients) |
| **Mertz 2004** [40] | Randomized Controlled trial | Yes | Double-blind | USA and Canada | Ribavirin | Placebo |
| **Vial 2013** [41] | Randomized Controlled trial | Yes | Double-blind | Chile | Methylprednisolone | Placebo |
| **Wernly 2011** [42] | Retrospective study | NA | NA | USA | Extracorporeal membrane oxygenation (ECMO) support | Patients intubated when they became hypoxic and placed on ECMO when they became hemodynamically unstable |
| **Vial 2015** [43] | Non-randomized comparative study | No | NA | Chile | Immune plasma infusion | No immune plasma infusion |
| **Hemorrhagic fever with renal syndrome** | | | | | | |

(*Continued*)

**Table 2.** (Continued)

| Study | Study design | Randomisation | Blinding | Country | Treatment | Comparator |
|---|---|---|---|---|---|---|
| **Gui 1987** [44] | Randomized Controlled trial | Yes | Single-blind | China | Recombinant interferon α-2 + Supportive therapy | Placebo + Supportive therapy |
| **Du 2013** [45] | Retrospective study | No | NA | China | Renal Replacement Therapy (RRT) | No RRT |
| **Huggins 1991** [46] | Randomized Controlled trial | Yes | Double-blind | China | Ribavirin | Placebo |
| **Lassa fever** | | | | | | |
| **Ilori 2019** [47] | Descriptive report | No | NA | Nigeria | Ribavirin + Supportive therapy | Supportive therapy |
| **McCormick 1986** [48] | Randomized Controlled trial | Yes | No | Sierra Leone | PHASE 1: No therapy VS Oral Ribavirin VS Lassa convalescent plasma | No therapy |
| | | | | | PHASE 2: intravenous ribavirin +/- lassa plasma | |

NA, Not applicable; NR, Not reported

## Ebola virus disease (EVD)

Seventeen studies on treatments were identified for EVD: two randomized controlled trials, two non-randomized comparative studies, four single-arm trials, eight retrospective studies and one descriptive report. Among these studies, four reported positive results on mortality. The two first studies showed that early vitamin A supplementation or IFN-beta 1a therapy may be associated with reduced mortality compared with no vitamin A supplementation or historical control (Relative risk reduction of mortality with vitamin A supplementation within 48h = 0.77 [0.59 to 0.99]; 21-day survival based on Kaplan Meier for IFN-beta 1a therapy versus historical control: 67% versus 19%). However, those results are to be taken with caution based on the weak design of the studies (retrospective study [22,23] and single-arm trial [33]).

The third study was a retrospective study assessing favipiravir [25] and reported that the treatment was associated with prolonged survival compared with standard therapy alone (CFR = 44% versus 65%). However, this result was challenged by another retrospective study (adjusted odds ratio = 0.48 [0.20 to 1.01]) [38] and a single-arm trial (CFR = 54% versus 55%), [26] which reported no significant results of favipiravir on mortality.

The last positive study was a randomized controlled trial assessing three treatments, the antiviral remdesivir, and the antibody-based therapies Mab114 and REGN-EB3 against ZMapp. [37] The study showed that both MAb114 and REGN-EB3 were superior to ZMapp (difference between MAb114 and ZMapp = -14.6% [-27.2 to -1.7]; difference between REGN-EB3 and ZMapp = -17.8% [-28.9 to -2.9]; in reducing mortality from EVD (with stringent findings among patients presenting with high levels of viral load).

## Hantavirus pulmonary syndrome (HPS)

Five studies on treatments were identified for HPS: two randomized controlled trials, two non-randomized comparative studies, and one retrospective study. Only two studies reported weakly significant results. The first study was a retrospective study evaluating the impact on survival of extracorporeal membrane oxygenation (ECMO) support in patients with HPS refractory to medical treatment and the associated complications. [42] This study reported a lower mortality in patients who had elective insertion of vascular sheaths and were almost concurrently intubated and placed on ECMO when they decompensated (CFR = 20%) compared with patients intubated when they became hypoxic and placed on ECMO when they became

**Table 3. Mortality results associated with VHFs treatments reported in the included studies.**

| Study | Comparison | Treatment group–No. | Comparator group–No. | Time | Treatment group—Mortality rate (%) | Comparator group—Mortality rate (%) | p-value | Outcome measure | Outcome result | P-value |
|---|---|---|---|---|---|---|---|---|---|---|
| **Argentine hemorrhagic fever** | | | | | | | | | | |
| **Maiztegui 1979** [11] | Immune plasma VS normal plasma | 91 | 97 | NR | 1.1 | 16.5 | <0,01 | NR | NR | NR |
| **Crimean-Congo hemorrhagic fever** | | | | | | | | | | |
| **Elaldi 2009** [12] | Ribavirin VS no ribavirin | 126 | 92 | First 8 days | 7.1 | 11.9 | 0.283 | Hazard ratio | 10.33 (1.65–64.71) | NR |
| | | | | After 8 days (to 30 days) | - | - | - | Hazard ratio | 0.1 (0.01–8.80) | NR |
| **Dokuzoguz 2013** [13] | Ribavirin +/- corticosteroids VS no treatment | Ribavirin +/- corticosteroids: 235 | 46 | NR | 7.7 | 10.9 | NR | Adjusted odds ratio | 0.04 (0.004–0.48) | 0.01 |
| | Corticosteroids + Ribavirin VS no treatment | Corticosteroids + Ribavirin: 44 | 217 | NR | 20.5 | 4.1 | NR | Adjusted odds ratio | 0.22 (0.039–1.27) | 0.092 |
| **Cevik 2008** [14] | Ribavirin VS no ribavirin | 9 | 16 | NR | 55.5 | 43.7 | 0.571 | NR | NR | NR |
| **Yilmaz 2016** [15] | Ribavirin VS no ribavirin | 198 | 345 | NR | 7.6 | 8.4 | 0.733 | NR | NR | NR |
| **Izadi 2009** [16] | Patients receiving ribavirin after onset of bleeding VS before onset of bleeding or had no bleeding at all | 39 | 24 | NR | 35.9 | 8.3 | NR | Odds ratio | 6.2 (1.3–30.3) | 0.018 |
| | Patients receiving ribavirin after the 4th day of disease onset VS within the first 4 days of disease onset | 25 | 38 | NR | 40.0 | 15.8 | NR | Odds ratio | 3.6 (1.1–11.6) | 0.031 |
| **Tulek 2012** [17] | Ribavirin VS no ribavirin | 91 | 152 | NR | 1.1 | 5.3 | 0.096 | NR | NR | NR |
| **Koksal 2010** [18] | Ribavirin VS supportive therapy | 64 | 72 | NR | 6.3 | 5.6 | 0.86 | NR | NR | NR |
| **Mardani 2003** [19] | Ribavirin VS historical control | 69 | 12 | NR | 11.6 | 58.3 | NR | Relative risk in confirmed patients | 0.20 (0.09–0.45) | <0.001 |
| **Salehi 2013** [20] | Immune globulins + ribavirin VS ribavirin | 12 | 28 | Until one week after hospitalisation | 25 | 11 | 0.24 | NR | NR | NR |
| **Ebola Virus Disease** | | | | | | | | | | |
| **Aluisio 2019** [21] | Intravenous fluid (IVF) | 70 | 354 | 28 days | 58.7 | 55.1 | 0.583 | NR | NR | NR |
| **Aluisio 2019** [22]**; Aluisio 2019** [23] | Vitamin A supplementation VS no vitamin A supplementation | 330 | 94 | During Ebola Treatment Unit care | 55 | 71.9 | NR | Relative risk reduction of mortality with vitamin A treatment within 48 h | 0.77 (0.59–0.99) | 0.041 |
| **Aluisio 2020** [24] | Cephalosporin VS no treatment | 360 | 64 | 48h | 58.7 | 55.1 | 0.583 | NR | NR | NR |
| **Bai 2016** [25] | Favipiravir VS supportive care | 39 | 85 | 60 days | 43.6 | 64.7 | 0.027 | NR | NR | NR |

*(Continued)*

**Table 3.** (Continued)

| Study | Comparison | Treatment group–No. | Comparator group–No. | Time | Treatment group—Mortality rate (%) | Comparator group—Mortality rate (%) | p-value | Outcome measure | Outcome result | P-value |
|---|---|---|---|---|---|---|---|---|---|---|
| **Sissoko 2016** [26] | Favipiravir VS historical control | 111 | NA | On-trial | 54.05 | 55 | NR | NR | NR | NR |
| **Yam 2020** [27] | Multivitamin supplementation VS no multivitamin supplementation | 261 | 163 | 48h | 53.6 | 63.8 | NR | NR | NR | NR |
| **Dunning 2016** [28] | Brincidofovir | 4 | NA | 14 days | 100 | NA | NR | NR | NR | NR |
| **Gignoux 2015** [29] | No Antimalarial Drug Prescription VS Artemether-Lumefantrine | 194 | 63 | NR | 64.4 | 65.1 | NR | Unadjusted risk ratio | 1.01 (0.82–1.25) | 0.92 |
| | Artesunate-Amodiaquine VS Artemether-Lumefantrine | 71 | 194 | NR | 50.7 | 64.4 | NR | Unadjusted risk ratio | 0.79 (0.61–1.01) | 0.06 |
| **Garbern 2019** [30] | Artesunate-amodiaquine VS no treatment | 22 | 402 | NR | 45.5 | 72.7 | NR | Risk of death in the matched cohort | 0.63 (0.37–1.07) | 0.086 |
| **Dunning 2016** [31] | TKM-130803 | 14 | 1820 (individual-level data from patients with PCR-confirmed Ebola infection from the 2014–2015 outbreak) | 14 days | 79 | 55 | NR | NR | NR | NR |
| **Sahr 2017** [32] | Convalescent whole blood VS routine care | 44 | 25 | NR | 27.9 | 44 | NR | Odds ratio for survival | 2.3 (0.8–6.5) | |
| **Konde 2017** [33] | IFN-beta 1a VS historical control | 9 | 38 | 21 days | 33.3 | 81 | NR | 21-day survival based on Kaplan Meier | 67% in treated patients VS 19% for the control | 0.026 |
| **Sadek 1999** [34] | Whole blood transfusion from convalescent patients | 8 | NA | NR | 12.5 | NR | NR | NR | NR | NR |
| **Van Griensven 2016** [35] | Convalescent plasma VS no treatment | 84 | 418 | 16 days | 31 | 38 | NR | Risk difference | -7% (-18 to 4) | NR |
| **Davey 2016** [36] | ZMapp VS standard of care | 35 | 36 | 28 days | 22 | 37 | NR | Absolute difference | -15% (-37 to 7) | NR |
| **Mulangu 2019** [37] | •Remdesivir VS ZMapp | 175 | 169 | 28 days | 53.1 | 49.7 | NR | Difference between groups | 3.4% (-7.2 to 14) | NR |
| | •Mab114 VS ZMapp | 174 | 169 | 28 days | 35.1 | 49.7 | NR | Difference between groups | -14.6% (-27.2 to -1.7) | 0.007 |

(*Continued*)

**Table 3.** (Continued)

| Study | Comparison | Treatment group–No. | Comparator group–No. | Time | Treatment group—Mortality rate (%) | Comparator group—Mortality rate (%) | p-value | Outcome measure | Outcome result | P-value |
|---|---|---|---|---|---|---|---|---|---|---|
| | •REGN-EB3 VS ZMapp | 155 | 169 | 28 days | 33.5 | 49.7 | NR | Difference between groups | -17.8% (-28.9 to -2.9) | 0.002 |
| **Kerber 2019** [38] | Favipiravir VS no treatment | 72 | 90 | NR | 42.5 | 57.8 | 0.053 | Adjusted odds ratio | 0.48 (0.20–1.01) | 0.11 |
| **Hantavirus Pulmonary Syndrome** | | | | | | | | | | |
| **Chapman 1999** [39] | Ribavirin VS no treatment | 30 | 34 | 1 year | 46.7 | 50 | NR | NR | NR | NR |
| **Mertz 2004** [40] | Ribavirin VS placebo | 10 | 13 | 28 days | 20 | 15 | 1 | NR | NR | NR |
| **Vial 2013** [41] | Methylprednisolone VS placebo | 30 | 30 | 28 days | 27 | 40 | 0.41 | Relative risk | 0.67 (0.32–1.39) | NR |
| **Wernly 2011** [42] | Patients who had elective insertion of vascular sheaths and were almost concurrently intubated and placed on ECMO when they decompensated VS patients intubated when they became hypoxic and placed on ECMO when they became hemodynamically unstable. | 25 | 26 | NR | 20 | 46 | 0.048 | NR | NR | NR |
| **Vial 2015** [43] | Immune plasma infusion VS no treatment | 29 | 199 | 30 days | 14 | 32 | 0.049 | Odds ratio | 0.35 (0.12–0.99) | NR |
| **Hemorrhagic fever with renal syndrome** | | | | | | | | | | |
| **Gui 1987** [44] | Recombinant interferon α-2 VS placebo | 25 | 25 | On-trial | 16 | 16 | NS | NR | NR | NR |
| **Du 2013** [45] | RRT VS no RRT | Total (both groups): 77 | NA | NR | 34.3 | 70 | 0.031 | NR | NR | NR |
| **Huggins 1991** [46] | Ribavirin VS placebo | 126 | 117 | On-trial | 2.4 | 8.55 | 0.01 | NR | NR | NR |
| **Lassa fever** | | | | | | | | | | |
| **Ilori 2019** [47] | Ribavirin VS no treatment | 334 | 21 | During the outbreak | 20.7 | 71.4 | <0.001 | NR | NR | NR |
| **McCormick 1986** [48] | Group of AST level>150: IV ribavirin VS no therapy | 63 | 60 | NR | 19 | 55 | 0.00003 | NR | NR | NR |
| | Group of AST level>150: Oral ribavirin VS no therapy | 14 | 60 | NR | 14 | 55 | 0.006 | NR | NR | NR |
| | Group of AST level>150: Plasma VS no therapy | 28 | 60 | NR | 50 | 55 | 0.3 | NR | NR | NR |

(*Continued*)

**Table 3.** (Continued)

| Study | Comparison | Treatment group–No. | Comparator group–No. | Time | Treatment group—Mortality rate (%) | Comparator group—Mortality rate (%) | p-value | Outcome measure | Outcome result | P-value |
|---|---|---|---|---|---|---|---|---|---|---|
| | Group of patients with virus level superior 10^3.6 TCID50/mL: IV ribavirin VS no therapy | 31 | 46 | NR | 32 | 76 | 0.00015 | NR | NR | NR |
| | Group of patients with virus level superior 10^3.6 TCID50/mL: Oral ribavirin VS no therapy | 10 | 46 | NR | 30 | 76 | 0.008 | NR | NR | NR |
| | Group of patients with virus level superior 10^3.6 TCID50/mL: Plasma VS no therapy | 21 | 46 | NR | 57 | 76 | 0.12 | NR | NR | NR |
| | Group of patients with virus level inferior 10^3.6 TCID50/mL: IV ribavirin VS no therapy | 32 | 111 | NR | 9 | 28 | 0.02 | NR | NR | NR |
| | Group of patients with virus level inferior 10^3.6 TCID50/mL: Oral ribavirin VS no therapy | 29 | 111 | NR | 7 | 28 | 0.01 | NR | NR | NR |
| | Group of patients with virus level inferior 10^3.6 TCID50/mL: Plasma VS no therapy | 32 | 111 | NR | 16 | 28 | 0.12 | NR | NR | NR |

NA, Not applicable; NR, Not reported

hemodynamically unstable (CFR = 46%). The second study was a non-randomized comparative study, which compared immune plasma infusion versus no treatment. [43] The study reported a weakly significant decrease in mortality associated with immune plasma infusion (Odds ratio = 0.35 [0.12 to 0.99]).

Therefore, no strong proof of effectiveness of treatment against HPS was identified.

## Hemorrhagic fever with renal syndrome (HFRS)

Three studies on treatments were identified for HFRS: two randomized controlled trials and one retrospective study. Only two studies reported positive results. The first study was a retrospective study comparing Renal Replacement Therapy (RRT) versus no RRT [45] and showed that RRT is associated with a decrease in mortality (CFR = 34% versus 70%). However, those results are to be taken with caution based on the weak design of the study. The second study was a randomized controlled trial assessing ribavirin [46] and showed a significant reduction in mortality among patients treated with ribavirin (CFR = 2% versus 9%).

## Lassa fever (LF)

Two studies on treatments were identified for LF: a randomized controlled trial and a descriptive report. The first study published in 1986 was a randomized controlled trial and evaluated

ribavirin and convalescent plasma compared with no therapy in several subgroups [48]. The study showed that ribavirin was associated with a significantly lower mortality than no therapy (CFR = 21% versus 71%). Moreover, the second study which was a descriptive report published in 2019, [47] also reported that ribavirin was associated with a decrease in mortality.

## Discussion

To our knowledge, this is the first comprehensive systematic review to summarize all published information available on worldwide cases numbers, mortality and treatments of a range of VHFs, excluding severe dengue and yellow fever.

Only a few number VHF systematic reviews were previously published and focused on single VHF, mainly EVD, CCHF and LF. Some meta-analyses were also conducted but often associated with heterogeneity issues. A previous meta-analysis on EVD found a pooled CFR of 60% in Africa [49]. However, this result was associated with a very high level of heterogeneity. This is consistent with our findings, which showed that CFR varied greatly across countries in outbreaks since 2010 ranging from 28% (2014–2016 outbreak in Sierra Leone) to 75% (2014 in Mali). In terms of specific treatments, a recent systematic review [50] focused mainly on the randomized controlled trial assessing remdesivir, Mab114 and REGN-EB3 against ZMapp. [37] The authors concluded as well that both MAb114 and REGN-EB3 were superior to ZMapp in reducing mortality from EVD with differences depending on the viral load at baseline.

Regarding CCHF, we identified in our review no strong proof of effectiveness of treatment. A previous meta-analysis on the efficacy of ribavirin in CCHF patients showed that ribavirin decreased the mortality rate compared with patients not treated with ribavirin.[51] However, this meta-analysis included an important number of low-quality studies such as case series. Therefore, the results should be considered with caution. Regarding Lassa fever, a previous meta-analysis on the efficacy of ribavirin showed that ribavirin was associated with lower risk of death than patients not treated with ribavirin.[52] However, heterogeneity was identified across studies and the results are mainly based on retrospective studies.

Our systematic reviews also has some limitations. One limitation of our review on the number of cases and CFR identified is that our findings are based on numbers registered on the WHO and CDC websites or published numbers, which can underestimate the reality. We also decided, when available, to prioritize laboratory confirmed cases numbers. Moreover, we restricted our review of the grey literature to national health websites and references reported on the WHO and CDC websites; data on clinical trials registries (e.g. ClinicalTrials.gov) were not included here. It is also important to mention that we did not stratify our results according to species or the strain of the virus. For example, in the case of Ebola disease investigational therapeutic options, the trials of importance were conducted during the 2014–2016 western Africa outbreak and the 2018–2020 North-Kivu (Democratic Republic of the Congo) outbreak that were related to Ebola virus (species Zaire Ebolavirus) and Makona and Kikwit specific strain respectively.[36,37] With respect to treatment of other ebolavirus diseases (e.g. Soudan and Bundibugyo virus diseases), options are even more limited. Notably, the three monoclonal antibody treatments tested in the PALM trial [37], ZMapp, REGN-EB3, and MAb114, have a narrow spectrum and are ineffective against other filovirus infections. In the case of other VHFs such as HPS or LF, the sparse comparative trials assessed mainly a nucleoside inhibitor (i.e., ribavirin). No strong proof of effectiveness of ribavirin was identified for HPS. Regarding LF, the efficacy of ribavirin was not considered as specific to Lassa virus lineage. Besides, considerable uncertainty was recently even more raised about its activity as an anti-infectious agent in the management of the condition.[52–54] Another limitation of our review is that we focused our results on the mortality of the VHF. However, the mortality does not account for

transmissibility of the virus, contagiousness and immune escape. Mortality rates varied also greatly across countries, especially in Africa, which can be explained by the fact that treatment facilities in some places in Africa may be limited. The study context also has an impact on the results of our review. Some authors report the challenge they can face of studying a relatively rare disease that affect widely dispersed rural areas.[41] Regarding VHFs such as EVD, authors reported difficulties during the course of their study, with for example interruption of participating centers due to violence from local community or paramilitary groups who can be suspicious of the activities in those facilities.[37] Moreover, a large number of included studies on treatment evaluation in our review were associated with a high risk of bias. This highlights the need to conduct clinical trials with a methodologically strong design.

This is of most importance to adapt the methodology of clinical trials in the specific context of VHFs. A recent study explored the application of Bayesian Decision Analysis (BDA) in order to incorporate the burden of disease and disease context into clinical trials, especially for the deadliest diseases in the US, such as cancers or liver cirrhosis. [55,56] This framework allows taking into account the disease context when determining the sample size and critical value of a fixed-sample test. Therefore, an interesting next step of our review would be to use these results to conduct a BDA to evaluate the optimal sample sizes and type I errors for future VHF clinical trials.

## Conclusion

We observed that number of cases and mortality varied greatly across VHFs as well as across countries within each VHF. The number of studies on VHFs treatments was very limited with very few randomized trials and no strong proof of effectiveness of treatment against most of the VHFs. Therefore, there is a high need of methodologically strong clinical trials conducted in the context of VHF.

## Supporting information

**S1 Fig. MEDLINE Search strategy.**
(DOCX)

**S1 Table. References details of each included publication.**
(DOCX)

**S2 Table. References details of each included WHO or CDC report.**
(DOCX)

**S3 Table. Quality Assessment of studies reporting results on treatments–Randomized Trials.**
(DOCX)

**S4 Table. Quality Assessment of studies reporting results on treatments–Non-Randomized Trials**
(DOCX)

**S5 Table. Quality Assessment of studies reporting number of cases and/or case fatality rates**
(DOCX)

**S6 Table. Number of cases and CFRs of Crimean-Congo hemorrhagic fever by country (alphabetical order) and period**
(DOCX)

**S7 Table. Number of cases and CFRs of Ebola Virus Disease by country and period**
(DOCX)

**S8 Table. Number of cases and CFRs of Hantavirus Pulmonary Syndrome by country and period**
(DOCX)

**S9 Table. Number of cases and CFRs of Hemorrhagic fever with renal syndrome by country and period**
(DOCX)

**S10 Table. Number of cases and CFRs of Lassa fever by country and period**
(DOCX)

**S11 Table. Number of cases and CFRs of Marburg virus disease by country and period**
(DOCX)

**S12 Table. Number of cases and CFRs of Rift Valley fever by country and period**
(DOCX)

**S13 Table. Number of cases and CFRs of Alkhurma hemorrhagic fever, Argentine hemorrhagic fever, Bolivian hemorrhagic fever, Chapare hemorrhagic fever, Lujo hemorrhagic fever, Omsk hemorrhagic fever, Sabia hemorrhagic fever and Venezuelan hemorrhagic fever by country and period**
(DOCX)

**S14 Table. Description of the treatment and comparator groups of each included studies reporting results on VHF treatments**
(DOCX)

## Author Contributions

**Conceptualization:** Drifa Belhadi, France Mentré, Cédric Laouénan.

**Data curation:** Drifa Belhadi.

**Formal analysis:** Drifa Belhadi, Majda El Baied, Guillaume Mulier.

**Methodology:** Drifa Belhadi.

**Project administration:** Drifa Belhadi, France Mentré, Cédric Laouénan.

**Supervision:** France Mentré, Cédric Laouénan.

**Validation:** Cédric Laouénan.

**Visualization:** Drifa Belhadi.

**Writing – original draft:** Drifa Belhadi.

**Writing – review & editing:** Drifa Belhadi, Majda El Baied, Guillaume Mulier, Denis Malvy, France Mentré, Cédric Laouénan.

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
