## [Decision Letter · Decision Letter 0]

13 Jul 2022

Dear Miss Belhadi,

Thank you very much for submitting your manuscript "The number of cases, mortality and treatments of viral hemorrhagic fevers: a systematic review" for consideration at PLOS Neglected Tropical Diseases. As with all papers reviewed by the journal, your manuscript was reviewed by members of the editorial board and by several independent reviewers. The reviewers appreciated the attention to an important topic. Based on the reviews, we are likely to accept this manuscript for publication, providing that you modify the manuscript according to the review recommendations. 

Please make sure that you address ALL the reviewers comments and recommendations. Please review the figures, including Fig 1 (Flowchart) making sure that the numbers add up and including the number of duplicates removed at each stage for each group. For example: 57+96=153 =! 141 in Panel A. Please also explain or clarify whether the Case fatality rate (CFR) was estimated by the authors of primary studies or if was instead estimated by the authors of this review using the derived information and how.

Sincerely,

Mabel Carabali, M.D., M.Sc., Ph.D.,

Academic Editor

Jeremy V. Camp, PhD

Section Editor

Please make sure that you address ALL the reviewers comments and recommendations. Please review the figures, including Fig 1 (Flowchart) making sure that the numbers add up and including the number of duplicates removed at each stage for each group. For example: 57+96=153 =! 141 in Panel A. Please also explain or clarify whether the Case fatality rate (CFR) was estimated by the authors of primary studies or if was instead estimated by the authors of this review using the derived information and how.

Reviewer's Responses to Questions

**Key Review Criteria Required for Acceptance?**

**Methods**

-Are the objectives of the study clearly articulated with a clear testable hypothesis stated?

-Is the study design appropriate to address the stated objectives?

-Is the population clearly described and appropriate for the hypothesis being tested?

-Is the sample size sufficient to ensure adequate power to address the hypothesis being tested?

-Were correct statistical analysis used to support conclusions?

-Are there concerns about ethical or regulatory requirements being met?

Reviewer #1: Methodology:

-Were the references of the articles included reviewed?

- The researchers reviewed review articles and their references, for example:

N Engl J Med 2020; 382:1832-1842

Reviewer #2: The methods have been clearly explained.

**Results**

-Does the analysis presented match the analysis plan?

-Are the results clearly and completely presented?

-Are the figures (Tables, Images) of sufficient quality for clarity?

Reviewer #1: Results:

Line 205 - Place the abbreviations on line 102

Line 374 – remind the reader that dengue and yellow fever were excluded

Tables:

-Table 2 – In the “Comparator” column adjust the descriptions. For example, one box says 'No ribavirin' (Cevik 2008, Yilmaz 2016) and another box says 'Only supportive therapy without ribavirin' (Koksal 2010). Does this mean that those who did not receive ribavirin also did not receive supportive treatment?

Is “NR” (Izadi 2009) the same as “No treatment” (Dokuzoguz 2013) or “No vitamin A supplementation” (Aluisio 2019 -408, 313)?

Same question for “No antimalarial drug prescription, Not exposed to ASAQ, Not treated with CWB, NA, etc.

-Clarify abbreviations: NR, NA (always place them at the foot of the tables)

Figures:

Check the resolution quality of the figures

Reviewer #2: The authors have done well in presenting their results. 

The author has taken an onerous task of carrying analyses and summarizing many studies on diseases occurring in such diverse circumstances. Despite being informative It risks diluting or oversimplifying results of such a heterogeneous group of diseases taking places in difference eras and circumstances. For example, Line 191: Did the authors look at whether there was a difference in the quality of studies between studies carried out in earlier years compared to later years? That should be mentioned. Other factors such as CFR can also be affected by the improvement in technology used as supportive in addition to medicinal treatments.

**Conclusions**

-Are the conclusions supported by the data presented?

-Are the limitations of analysis clearly described?

-Do the authors discuss how these data can be helpful to advance our understanding of the topic under study?

-Is public health relevance addressed?

Reviewer #1: The conclusions are clear

Reviewer #2: It should be noted in the conclusion that treatment facilities in some places in Africa where treatment for VHF cases was taking place in the during period of outbreak, may not be idea and can account for some higher CFR.

It is important for the authors to make reference to some important inherent weaknesses of the study context as described by the authors of each study that has been included in the analysis.

**Editorial and Data Presentation Modifications?**

Reviewer #1: Minor Revision

Reviewer #2: Accept

**Summary and General Comments**

Reviewer #1: This study aimed to carry out a systematic review of all reports and clinical studies that included specific results on the number of cases, mortality, and treatments of viral hemorrhagic fevers (VHF), focused mainly on BSL-4 pathogens (excluded dengue and yellow fever). 

Results from the last two decades are presented, where limited therapeutic options are observed.

This results in a call to action for decision-makers to be aware of the existing gap and propose new studies.

Advantage:

-The revision is registered in PROSPERO

Disadvantages/Limitations:

-The databases included were MEDLINE, Embase, and CENTRAL

-Gray literature (e.g., OpenGrey) and data on ClinicalTrials.gov were not included

-Limited to English and French language (add this information in the methods section)

***Consider adding these limitations in the discussion section (line 393)

Reviewer #2: (No Response)

PLOS authors have the option to publish the peer review history of their article (what does this mean?). If published, this will include your full peer review and any attached files.

Reviewer #1: Yes: Luis Gabriel Parra-Lara

Reviewer #2: No

Figure Files:

Data Requirements:

Reproducibility:

References

---

## [Decision Letter · Decision Letter 1]

28 Sep 2022

Dear Miss Belhadi,

Thank you very much for submitting your manuscript "The number of cases, mortality and treatments of viral hemorrhagic fevers: a systematic review" for consideration at PLOS Neglected Tropical Diseases. As with all papers reviewed by the journal, your manuscript was reviewed by members of the editorial board and by several independent reviewers. The reviewers appreciated the attention to an important topic. Based on the reviews, we are likely to accept this manuscript for publication, providing that you modify the manuscript according to the review recommendations. 

Although the manuscript went through a first round of review, we asked additional reviewer to go through the manuscript given some outstanding relevant inconsistencies. Before considering the manuscript for publication, it would be necessary for the authors to address ALL the comments indicated by the reviewers.

Sincerely,

Mabel Carabali, M.D., M.Sc., Ph.D.,

Academic Editor

Jeremy Camp

Section Editor

Although the manuscript went through a first round of review, we asked additional reviewer to go through the manuscript given some outstanding relevant inconsistencies. Before considering the manuscript for publication, it would be necessary for the authors to address ALL the comments indicated by the reviewers.

Reviewer's Responses to Questions

**Key Review Criteria Required for Acceptance?**

**Methods**

-Are the objectives of the study clearly articulated with a clear testable hypothesis stated?

-Is the study design appropriate to address the stated objectives?

-Is the population clearly described and appropriate for the hypothesis being tested?

-Is the sample size sufficient to ensure adequate power to address the hypothesis being tested?

-Were correct statistical analysis used to support conclusions?

-Are there concerns about ethical or regulatory requirements being met?

Reviewer #1: (No Response)

Reviewer #3: see below

**Results**

-Does the analysis presented match the analysis plan?

-Are the results clearly and completely presented?

-Are the figures (Tables, Images) of sufficient quality for clarity?

Reviewer #1: (No Response)

Reviewer #3: see below

**Conclusions**

-Are the conclusions supported by the data presented?

-Are the limitations of analysis clearly described?

-Do the authors discuss how these data can be helpful to advance our understanding of the topic under study?

-Is public health relevance addressed?

Reviewer #1: (No Response)

Reviewer #3: see below

**Editorial and Data Presentation Modifications?**

Reviewer #1: (No Response)

Reviewer #3: see below

**Summary and General Comments**

Reviewer #1: (No Response)

Reviewer #3: Note: I was asked to review this manuscript after it had already been through a round of peer review. 

The stated goal of the manuscript was to review CFR, geography and treatment of "VHFs". There is a lack of precision in the term "VHF". The authors exclude two of the most globally relevant VHFs: YF and Dengue, yet include new world hantaviruses and LCMV, which are associated with pulmonary syndrome and encephalitis respectively and don't exhibit prominent hemorrhagic symptoms. They also state that they are looking at "BSL-4" VHFs but the Hantaviruses, LCMV and RVFV are NOT BSL-4 viruses. Some clarity is needed regarding which pathogens are selected for study and why. Maybe since there really wasn't any data for the tickborne flavi's, you just exclude them and state that you are looking at selected bunyas, filos and arenas? this is c/w the ones you chose for narrative description and figures.

Table 3: I assume HR is Hazard Ratio, but this is not defined in the table. How can a study have an outcome result and a p value when the outcome has no measure? eg Dokuzokguz 2013 corticosteroids +/- ribavirin vs no treatment- all parameters are listed as NR (ie not reported, according to the table).

Dunning 2016-1888 (what does 1888 here mean?): how can you have an outcome in the comparator group but no N in the comparator group?

Some mention should be made, especially for the Ebola treatment data that it only applies to one strain of virus (ie Zaire, 

 rather than Sudan, Bundi, etc). Same for HPS- are these studies from patients with Andes? SNV? a different New World Hanta? Lassa from different regions also has very divergent phylogeny, so results are not generalizable to all disease.

PLOS authors have the option to publish the peer review history of their article (what does this mean?). If published, this will include your full peer review and any attached files.

Reviewer #1: Yes: Luis Gabriel Parra-Lara

Reviewer #3: No

Figure Files:

Data Requirements:

Reproducibility:

References

---

## [Editor Report · Decision Letter 2]

14 Oct 2022

Dear Miss Belhadi,

We are pleased to inform you that your manuscript 'The number of cases, mortality and treatments of viral hemorrhagic fevers: a systematic review' has been provisionally accepted for publication in PLOS Neglected Tropical Diseases.

Best regards,

Mabel Carabali, M.D., M.Sc., Ph.D.,

Academic Editor

Jeremy V. Camp, PhD

Section Editor

The authors addressed all comments indicated by the reviewers.

---

## [Editor Report · Acceptance letter]

28 Oct 2022

Dear Miss Belhadi,

We are delighted to inform you that your manuscript, "The number of cases, mortality and treatments of viral hemorrhagic fevers: a systematic review," has been formally accepted for publication in PLOS Neglected Tropical Diseases.

Best regards,

Shaden Kamhawi

co-Editor-in-Chief

Paul Brindley

co-Editor-in-Chief
